# Determinants of births protected against neonatal tetanus in Ethiopia: A multilevel analysis using EDHS 2016 data

Achamyeleh Birhanu Teshale *, Getayeneh Antehunegn Tesema

Department of Epidemiology and Biostatistics, Institute of Public Health, College of Medicine and Health Sciences, University of Gondar, Gondar, Ethiopia

* achambir08@gmail.com

## Abstract

### Background

Even though there is low coverage of maternal health services such as antenatal care and skilled birth attendant delivery as well as poor sanitary practice during delivery in Ethiopia, the proportion of births protected by the tetanus vaccine is low. Thus, this study aimed to investigate the determinants of births protected against neonatal tetanus in Ethiopia.

### Objective

To assess the determinants of births protected against neonatal tetanus in Ethiopia.

### Method

The study was based on secondary data analysis of the Ethiopian Demographic and Health Survey 2016 data. A weighted sample of 7590 women who gave birth within five years preceding the survey was used for analysis. We conducted a multilevel analysis, due to the hierarchical nature of the data. Variables with p-value <0.05 in the multivariable analysis were declared to be significantly associated with having births protected against neonatal tetanus.

### Result

In this study, mothers with primary education [adjusted odds ratio (AOR) = 1.23; 95%CI: 1.04, 1.44] and secondary and above education [AOR = 1.36; 95%CI: 1.06, 1.73], media exposure [AOR = 1.35; 95%CI: 1.15, 1.58], not perceiving distance from the health facility as a big problem [AOR = 1.24; 95%CI: 1.08,1.42], one antenatal care (ANC) visit [AOR = 1.56; 95%CI: 2.71, 4.68], two to three ANC visit [AOR = 11.82; 95%CI: 9.94,14.06], and four and more ANC visit [AOR = 15.25; 95%CI: 12.74, 18.26], being in Amhara [AOR = 0.59; 95%CI: 0.38,0.92], Afar [AO = 0.41; 95%CI: 0.25,0.66], and Harari [AOR = 1.88; 95%CI: 1.15,3.07] regions, being in communities with higher level of women education [AOR = 1.25; 95%CI: 1.03,1.52], and higher level of media exposure [AOR = 1.22; 95%CI: 1.01,1.48] were significant predictors of having a protected birth against neonatal tetanus.

**Data Availability Statement:** The Ethiopian Demographic and Health Survey 2016 data can be accessed from the Measure DHS program at www.dhsprogram.com, through legal requesting.

**Funding:** The authors received no specific funding for this work.

**Competing interests:** The authors have declared that no competing interests exist.

**Abbreviations:** ANC, Antenatal Care; AOR, Adjusted Odds ratio; EAs, Enumeration Areas; EDHS, Ethiopian Demographic and Health Survey; ICC, Intraclass Correlation Coefficient; MNT, Maternal and Neonatal Tetanus; MOR, Median odds ratio; NT, Neonatal Tetanus; PCV, Proportional Change in Variance; PHC, Ethiopia Population and Housing Census; SNNPR, Southern Nation Nationalities and People's Region; TT, tetanus Toxoid.

## Conclusion

In this study, both individual level and community level factors were associated with having protected birth against neonatal tetanus. Therefore, strengthening maternal health services such as ANC visits and interventions related to increasing media campaigns regarding tetanus could increase the immunization against tetanus among reproductive-age women. In addition, it is also better to give attention to those reproductive age group women from remote areas and also better to distribute maternal services fairly and equally between regions.

## Background

In Ethiopian, the Expanded Program on Immunization (EPI) was launched in 1980 [1]. Its target groups are children under one year of age and women of reproductive age group (15–49 years of age). The currently available EPI vaccines in Ethiopia are Bacille Calmette-Guerin (BCG), Measles, DPT-HepB-Hib or pentavalent, Rotavirus, Pneumococcus Vaccine (PCV), Oral Polio Vaccine (OPV), and Tetanus Toxoid (TT) Vaccines [1].

Tetanus is one of the vaccine-preventable bacterial disease caused by a toxin produced by Clostridium tetani [2]. Maternal and neonatal tetanus (MNT) is a triple failure of the public health system (immunization Programme, antenatal care, and clean and safe birth practices) which mostly affect disadvantaged and underserved population groups, who did not have access to adequate health services [3]. Because most neonatal tetanus (NT) infections occur during childbirth, due to inadequate/poor sanitary conditions, newborns need to have maternal antibodies against tetanus that are obtained through the placenta [3,4].

Even though the elimination of MNT was achieved in most of the countries, by the end of 2015, there were still 21 countries that had not yet attained the elimination of MNT [3]. Globally by 2017, there were 30,848 newborn deaths due to neonatal tetanus [5]. In many lower and middle-income countries, in which many mothers and neonates died at home during delivery, births and deaths are not officially reported and the burden of mortality due to tetanus cannot be estimated [3]. Despite Ethiopia validated the elimination of MNT, it is not an end in itself and its maintenance needs an ongoing vaccination Programme and improved public health infrastructure [6].

Giving tetanus immunization for women of childbearing age and pregnant women is an intervention that is taken to protect both the mother and the newborns from tetanus [7]. If mothers took the appropriate number of doses of the TT vaccine during or before pregnancy, both the mother and her child will be safe from tetanus during delivery [8].

In many countries including Ethiopia, TT vaccination is part of routine maternal health care services in which a minimum of two doses of the vaccine is given to pregnant women during pregnancy if the mother did not taka the vaccine before. However, the overall childbearing mothers should take five doses of TT vaccine. These five doses protect tetanus throughout women's reproductive years [3,9,10].

Globally 82% of newborns were protected at birth through maternal TT vaccination, with at least two doses of tetanus toxoid vaccine [11]. In Africa, the proportions of childbearing women with at least two doses of tetanus-containing vaccine and the proportions of newborns protected at birth were 69% and 77% respectively in 2015 [12]. In Ethiopia, according to the Ethiopian Demographic and Health Survey (EDHS) 2016 report, only 49% of women had their last birth protected against neonatal tetanus/had received sufficient doses of TT vaccine [13].

Evidences revealed different determinants of births protected against NT such as maternal age [14], maternal education [15–18], maternal occupation [17,19], marital status [17,20], wealth status [19–21], birth order [19], antenatal care (ANC) [19,22,23], distance from the health facility [16,21,23], media exposure [22,24–26], residence [14], and region [24,27,28].

Even though there is low coverage of maternal health services such as ANC (only 32% of mothers had at least 4 ANC visits) and skilled birth attendant delivery (only 26% of mothers delivered in the presence of skilled birth attendant) as well as poor sanitary practice during delivery in Ethiopia [13], the proportion of births protected against NT is low. Besides, to the best of our knowledge evidence showing the factors influencing births protected against NT is limited in Ethiopia and no study was done based on nationally representative data. In addition, previous studies consider only individual level factors while our study considers both individual and community level determinants of births protected against NT. Thus, this study aimed to investigate the determinants of births protected against neonatal tetanus in Ethiopia. The finding of this study would probably help health professionals and policymakers to generate evidence that strengthens and maintain current efforts of eliminating MNT.

## Method

### Data source and study population

This study was based on secondary data analysis of the 2016 EDHS, which was conducted from January 18, 2016, to June 27, 2016. The sampling frame used for the 2016 EDHS was a complete list of 84,915 enumeration areas (EAs) created for the 2007 Population and Housing Census (PHC). The survey used a stratified cluster sampling selected in two-stages. In the first stage, a total of 645 clusters or EAs were selected and in the second stage, 28 households per cluster were selected. For our study, a total weighted sample of 7590 women who gave birth within five years preceding the survey was used. Detailed information on sampling technique and questioner, in general about the survey, is obtained from the EDHS 2016 report [13].

### Variables of the study

The outcome variable was a birth protected against neonatal tetanus, which was a binary outcome variable coded as "0" if it was not protected and "1" if it was protected. The independent variables in this study were further classified into individual and community level factors. The individual-level factors used in this study were; maternal age, maternal education, maternal occupation, marital status, household wealth status, media exposure, perception of distance from the health facility, birth order, household size, wanted last-child, ANC visit for their last pregnancy, and ever had of a terminated pregnancy. Four community-level variables; residence, region, community-level media exposure, and community level of women's education were also used as an independent variable in this study. The community-level factors, community-level media exposure, and community level of women's education were created by aggregating individual-level factors since these variables are not directly found from the survey.

### Operational definitions

**Household wealth status.** Derived using principal components analysis and it was directly available in the EDHS dataset with the five categories (lowest, second, middle, fourth, and highest) [13]. It was re-coded as poor (includes the lowest and second category), middle, and rich (includes the fourth and richest categories) for our analysis

**Media exposure.**   Created by combining whether a respondent reads the newspaper, listens to the radio, and watch television and coded as "yes" if the mother was exposed to at least one of the three media and "no" otherwise.

**Community level of media exposure.**   A community level variable measured by the proportion of women who had exposed to at least one media; television, radio, or newspaper and categorized based on national median value as low (communities with <50% of women exposed) and high (communities with ≥50% of women exposed) community level media exposure.

**Community level of women education.**   Aggregate values measured by the proportion of women with a minimum of primary level of education derived from data on respondents' level of education. Then, it was categorized using national median value to values: low (communities with <50% of women have at least primary education) and high (communities with ≥ 50% of women have at least primary education) community level of women education.

## Data management and statistical analysis

Further coding of the data and analysis was done using Stata version 14. Throughout the study, weighting was done to adjust for non-proportional sample selection and for non-responses as well to restore the representativeness of the data. We used the multilevel logistic model, because of the EDHS data by itself is hierarchical. We first conducted bivariable multilevel logistic regression analysis and then we fitted multivariable multilevel logistic regression analysis, for variables with p<0.20 in the bivariable analysis. In the multivariable multilevel logistic regression analysis, variables with p<0.05 were declared to be significantly associated with having births protected against NT. We fitted four models containing variables of interest. The null model (fitted without explanatory variables), Model I (examined the effects of individual-level factors only), model II (containing only community-level factors), and Model III containing both individual and community-level factors. Intraclass correlation coefficient (ICC), a proportional change in variance (PCV), and median odds ratio (MOR) were used to examine clustering and the extent to which community-level factors explain the unexplained variance of the null model. Model comparison/fitness was checked by deviance and the model with the lowest deviance was used as the best-fitted model.

## Ethical consideration

The EDHS was conducted based on the permission of the government, and informed consent was taken and participants' confidentiality was assured during that time. For this study, we accessed the data set based upon request ([www.dhsprogram.com](www.dhsprogram.com) online) and there was no ethical approval required. Moreover, there are no names of individuals or household addresses in the data files.

## Results

### Sociodemographic characteristics

A total weighted sample of 7590 women who gave birth within five years preceding the survey was used in our analysis. The median age of the study participants was 28(IQR = 24–34) years. The majority (63.12%) of the study participants had no formal education and 43.55% were from poor wealth status. More than two thirds (66.21%) of women had no media exposure and 58.06% of women perceive distance from the health facility as a big problem. The majority (73.43%) of women's last child was wanted. Regarding birth order and ANC visit, 51.04% and 37.13% of study participants had a birth order of four and above and no ANC visit respectively.

Most (87.23%) of the respondents were from a rural part of Ethiopia. Regarding region, 41.23%, 21.50%, and 21.09% of study participants were from Oromia, Amhara, and Southern Nation Nationalities and Peoples Region (SNNPR) respectively. While small proportions of study participants were from Harari, Gambela, and Dire Dawa. More than half of the study participants were from communities with a higher level of women education and media exposure (Table 1).

**Table 1. Respondents sociodemographic characteristics.**

| Variables | Frequency | Percentage |
|---|---|---|
| Maternal age | | |
| 15–19 | 339 | 4.47 |
| 20–24 | 1,465 | 19.30 |
| 25–29 | 2,165 | 28.53 |
| 30–34 | 1,661 | 21.89 |
| 35–39 | 1,206 | 15.89 |
| 40–44 | 547 | 7.20 |
| 45–49 | 207 | 2.73 |
| Maternal education | | |
| No formal education | 4,791 | 63.12 |
| Primary education | 2,150 | 28.32 |
| Secondary and above | 649 | 8.55 |
| Maternal occupation | | |
| Working | 3,512 | 46.27 |
| Not working | 4,078 | 53.73 |
| Marital status | | |
| Married | 7,020 | 92.49 |
| Unmarried | 570 | 7.51 |
| Household size | | |
| Less than five | 3,636 | 47.92 |
| Five or above | 3,954 | 52.09 |
| Household wealth status | | |
| Poor | 3,306 | 43.55 |
| Middle | 1,588 | 20.93 |
| Rich | 2,696 | 35.52 |
| Media exposure | | |
| Yes | 2,565 | 33.79 |
| No | 5,025 | 66.21 |
| Perception of distance from the health facility | | |
| A big problem | 4,407 | 58.06 |
| Not a big problem | 3,183 | 41.94 |
| Birth order | | |
| 1st | 1,434 | 18.90 |
| 2nd to 3rd | 2,282 | 30.06 |
| 4th and above | 3,874 | 51.04 |
| Wanted the last child | | |
| Wanted | 5,574 | 73.43 |
| Not wanted | 2,016 | 26.57 |
| ANC visit | | |
| No ANC visits | 2,818 | 37.13 |

(*Continued*)

**Table 1.** (Continued)

| Variables | Frequency | Percentage |
|---|---|---|
| One | 335 | 4.41 |
| Two to three | 2,007 | 26.45 |
| Four and above | 2,430 | 32.01 |
| Ever had of a terminated pregnancy | | |
| Yes | 680 | 8.96 |
| No | 6,910 | 91.04 |
| Residence | | |
| Urban | 969 | 12.77 |
| Rural | 6,621 | 87.23 |
| Region | | |
| Tigray | 537 | 7.08 |
| Afar | 71 | 0.94 |
| Amhara | 1,632 | 21.50 |
| Oromia | 3,130 | 41.23 |
| Somalia | 269 | 3.54 |
| Benishangul | 81 | 1.06 |
| SNNPR | 1,601 | 21.09 |
| Gambela | 21 | 0.27 |
| Harari | 17 | 0.23 |
| Addis Ababa | 198 | 2.61 |
| Dire Dawa | 33 | 0.44 |
| Community-level of women education | | |
| Low | 3,744 | 49.33 |
| High | 3,846 | 50.67 |
| Community-level of media exposure | | |
| Low | 3,475 | 45.79 |
| High | 4,115 | 54.21 |

## Random effect model and model fitness

As shown in Table 2, the ICC in the null model was 0.307, which indicates about 30.7% of the variations in having births protected against NT were attributable to differences between clusters/communities. Similarly, the higher MOR value (3.17) in the null model indicates there was significant variation between clusters. This is interpreted as; if we randomly choose an individual from two different clusters, those from a higher risk cluster had 3.16 times higher odds of being having births protected against NT as compared to those individuals who come from the lower risk cluster. Furthermore, the higher PCV (77%) in the final model revealed that 77% of the variations of protected birth against neonatal tetanus were attributable to both

**Table 2. Random effect model and model fitness for determinants of births protected against neonatal tetanus.**

| Parameter | Null model | Model I | Model II | Model III |
|---|---|---|---|---|
| ICC | 0.307 | 0.129 | 0.149 | 0.094 |
| MOR | 3.16(2.86–3.53) | 1.93(1.79–2.12) | 2.05(1.90–2.25) | 1.75(1.60–1.92) |
| PCV | Reference | 0.67 | 0.61 | 0.77 |
| Model fitness | | | | |
| Deviance | 9106.8054 | 7346.89 | 8711.55 | 7257.66 |

individual and community-level factors. Table 2 also revealed the best-fitted model was the final model (model III) since it had the lowest deviance (7257.66).

## Determinants of births protected against neonatal tetanus

For multivariable multilevel analysis, we consider only variables with p<0.2 in the bivariable analysis. In the multivariable multilevel analysis maternal education, media exposure, perception of distance from the health facility, ANC visit, region, community-level women education, and community-level media exposure were significantly associated with having births protected against NT. The odds of having a protected birth against NT was 1.23 [adjusted odds ratio (AOR) = 1.23; 95%CI: 1.04, 1.44] and 1.36 [AOR = 1.36; 95%CI: 1.06, 1.73] times higher among mothers with primary education and secondary and above educational level respectively as compared to mothers who had no formal education. Mothers who were exposed to media had 1.35 [AOR = 1.35; 95%CI: 1.15, 1.58] times higher odds of having a protected birth against NT as compared to their counterparts. Looking at the perception of distance from the health facility, mothers who did not perceive distance from the health facility as a big problem had 1.24 [AOR = 1.24; 95%CI: 1.08,1.42] times higher odds of having a protected birth against NT as compared to their counterpart. The odds of having a protected birth against NT was 1.56 [AOR = 1.56; 95%CI: 2.71, 4.68], 11.82 [AOR = 11.82; 95%CI: 9.94,14.06], and 15.25 [AOR = 15.25; 95%CI: 12.74, 18.26] times among mothers who attend one, two to three and four and above ANC visits respectively as compared to mothers who had no ANC follow up. Regarding region, mothers from Amhara and Afar had 41% [AOR = 0.59; 95%CI: 0.38, 0.92] and 59% [AOR = 0.41; 95%CI: 0.25, 0.66] lower odds of having births protected against NT as compared to mothers from Addis Ababa. Besides, mothers from Harari had 1.88 [AOR = 1.88; 95%CI: 1.15, 3.07] times higher odds of having a protected birth against NT. The odds of having a protected birth against NT was 1.25 [AOR = 1.25; 95%CI: 1.03, 1.52] and 1.22 [AOR = 1.22; 95%CI: 1.01, 1.48] times among mothers from communities with higher-level women education and a higher level of media exposure respectively as compared to their counterpart (Table 3).

## Discussion

Even though in Ethiopia giving birth in unsanitary conditions is common, less than half of births are protected against NT [13]. Therefore, we investigated the determinants of births protected against neonatal tetanus in Ethiopia using EDHS 2016 data. Both individual level and community level factors were associated with having births protected against NT. Among the individual-level factors maternal education, media exposure, perception of distance from the health facility, and ANC visit were associated with having births protected against NT. Of community-level factors region, community level of women education, and community level of media exposure were significantly associated with having births protected against NT.

In this study mothers having primary, and secondary and above education were more likely to have births protected against NT as compared to those mothers with no formal education. This finding is in line with studies done in Egypt [15], Nigeria [17], and Ethiopia [18]. The possible explanation might be that educated mothers might generally have greater knowledge and awareness regarding the benefits of immunization. Besides, educated mothers might have a greater decision-making power regarding their health and they mostly have the freedom to travel outside the home to seek care which can improve uptake of immunizations such as immunization against tetanus.

Mothers with media exposure had higher odds of having births protected against NT as compared to their counterparts. This is congruent with studies done in sub-Saharan Africa

**Table 3. Multivariable multilevel logistic regression analysis of determinants of births protected against neonatal tetanus in Ethiopia, 2016.**

| Variables | Null model | Model I (AOR 95%CI) | Model II (AOR 95%CI) | Model III (AOR 95%CI) |
|---|---|---|---|---|
| **Maternal age** | | | | |
| 15–19 | | 1.00 | | 1.00 |
| 20–24 | | 1.25(0.92–1.69) | | 1.26(0.93–1.71) |
| 25–29 | | 1.14(0.83–1.58) | | 1.14(0.83–1.57) |
| 30–34 | | 1.22(0.86–1.72) | | 1.20(0.84–1.70) |
| 35–39 | | 1.06(0.74–1.53) | | 1.05(0.73–1.52) |
| 40–44 | | 0.96(0.63–1.45) | | 0.97(0.64–1.47) |
| 45–49 | | 1.26(0.75–2.13) | | 1.29(0.77–2.18) |
| **Maternal education** | | | | |
| No formal education | | 1.00 | | 1.00 |
| Primary education | | 1.32(1.13–1.54) | | 1.23(1.04–1.44) * |
| Secondary and above | | 1.50(1.18–1.89) | | 1.36(1.06–1.73) * |
| **Maternal occupation** | | | | |
| Working | | 1.06(0.93–1.20) | | 1.06(0.93–1.20) |
| Not working | | 1.00 | | 1.00 |
| **Household size** | | | | |
| Less than five | | 1.00 | | 1.00 |
| Five or above | | 1.07(0.92–1.25) | | 1.07(0.92–1.25) |
| **Household wealth status** | | | | |
| Poor | | 1.00 | | 1.00 |
| Middle | | 1.09(0.91–1.31) | | 1.02(0.84–1.22) |
| Rich | | 1.35(1.13–1.61) | | 1.16(0.96–1.40) |
| **Media exposure** | | | | |
| Yes | | 1.44(1.24–1.68) | | 1.35(1.15–1.58) *** |
| No | | 1.00 | | 1.00 |
| **Perception of distance from the health facility** | | | | |
| A big problem | | 1.00 | | 1.00 |
| Not a big problem | | 1.25(1.09–1.43) | | 1.24(1.08–1.42) ** |
| **Birth order** | | | | |
| 1st | | 1.00 | | 1.00 |
| 2nd to 3rd | | 1.11(0.92–1.35) | | 1.10(0.91–1.33) |
| 4th and above | | 1.28(0.99–1.65) | | 1.26(0.98–1.62) |
| **Wanted the last child** | | | | |
| Wanted | | 1.00 | | 1.00 |
| Not wanted | | 1.09(0.93–1.27) | | 1.08(0.92–1.26) |
| **ANC visit** | | | | |
| No ANC visits | | 1.00 | | 1.00 |
| One | | 3.57(2.72–4.69) | | 3.56(2.71–4.68) *** |
| Two to three | | 11.70(9.85–13.91) | | 11.82(9.94–14.06) *** |
| Four and above | | 15.30(12.83–18.24) | | 15.25(12.74–18.26) *** |
| **Ever had of a terminated pregnancy** | | | | |
| Yes | | 1.00 | | 1.00 |
| No | | 1.11(0.90–1.37) | | 1.13(0.91–1.39) |
| **Residence** | | | | |
| Urban | | | 1.00 | 1.00 |
| Rural | | | 0.58(0.45–0.75) | 1.15(0.89–1.51) |
| **Region** | | | | |

*(Continued)*

**Table 3.** (Continued)

| Variables | Null model | Model I (AOR 95%CI) | Model II (AOR 95%CI) | Model III (AOR 95%CI) |
|---|---|---|---|---|
| Addis Ababa | | | 1.00 | 1.00 |
| Tigray | | | 0.77(0.48–1.22) | 0.64(0.42–1.01) |
| Afar | | | 0.22(0.13–0.36) | 0.41(0.25–0.66) *** |
| Amhara | | | 0.54(0.34–0.86) | 0.59(0.38–0.92) * |
| Oromia | | | 0.52(0.33–0.83) | 1.08(0.70–1.68) |
| Somalia | | | 0.48(0.30–0.78) | 1.10(0.70–1.73) |
| Benishangul | | | 0.79(0.48–1.30) | 0.93(0.58–1.49) |
| SNNPR | | | 0.62(0.39–0.98) | 071(0.46–1.10) |
| Gambela | | | 0.46(0.28–0.74) | 0.76(0.48–1.20) |
| Harari | | | 1.19(0.72–1.98) | 1.88(1.15–3.07) * |
| Dire Dawa | | | 0.99(0.60–1.62) | 1.143(0.71–1.83) |
| **Community level of women education** | | | | |
| Low | | | 1.00 | 1.00 |
| High | | | 1.95(1.59–2.39) | 1.25(1.03–1.52) * |
| **Community-level of media exposure** | | | | |
| Low | | | 1.00 | 1.00 |
| High | | | 1.88(1.55–2.29) | 1.22(1.01–1.48) * |

**Note**; AOR = Adjusted Odds Ratio, CI = Confidence Interval

* = P<0.05

** = P<0.01

*** = P<0.001

[26], Nigeria [25], and Ethiopia [18]. This is because, in recent years information regarding maternal and child health are distributed through different medias such as television, radio, and newspapers and this might increase the mother's knowledge on safe motherhood and utilization of maternal health services. In addition, media exposure is helpful for the adoption of different behaviors that bring positive behavioral changes towards immunization for vaccine-preventable diseases.

Consistent with studies done in Ethiopia [18,23], mothers who perceive distance from the health facility as a big problem had lower odds of having births protected against NT. This might be due to the costs such as time and transportation due to distant/remote health centers or vaccination centers. In addition, since full tetanus immunization in reproductive age women needs repeated visits to health facilities, such extra visits might be boring and tiring for women especially if they are far from the institution, which gives the immunization service.

Antenatal care visit was another factor that was associated with having births protected against NT in which mothers who had ANC visits were more likely to have births protected against NT. This is congruent with studies in Kenya [19] and Ethiopia [18,23]. This might be because in Ethiopia tetanus immunization is one of the ANC service package and women who had ANC visits had an opportunity to take the immunization. The other possible explanation is contact with healthcare providers might allow getting information about the benefits of taking full or recommended doses of tetanus for both the mother and the newborn.

There were also regional variations of births protected against NT in this study. This regional variation of tetanus immunization is consistent with different studies conducted elsewhere [24,27,28]. This might be because of the sociocultural difference as well as due to inequality in the distribution of health facilities and health personnel in different regions of

Ethiopia. Moreover, mothers from border regions such as Afar might have limited access to information regarding maternal health services and other services like access to school/education.

Moreover, mothers from communities with a higher level of women education have higher odds of having births protected against NT. This might be due to communities with a high concentration of educated women indicates greater awareness, autonomy, and decision-making power for utilization of maternal health services during pregnancy and childbirth and this might intern can have its influence on taking vaccinations for vaccine-preventable diseases such as tetanus. In this study, we also found that mothers from communities with a higher level of media exposure have higher odds of having births protected against neonatal tetanus. This might be women from communities with a higher level of media exposure might get important and timely information about devastating but vaccine-preventable diseases such as tetanus.

This study had both strengths and limitations. Regarding the strength of the study, since it was based on the representative EDHS data, we can generalize our findings to the reproductive age group women in Ethiopia. Besides, we used an appropriate model (multilevel analysis) considering the hierarchical nature of the data and we can have a better estimation of parameters. However, in our study, important variables such as women's knowledge regarding tetanus was not assessed since this variable was not found in the survey. Since it was based on a maternal report for the question "how many tetanus doses (number of vaccinations) did you took during your lifetime and your last pregnancy" there may be recall bias. Moreover, we cannot establish the temporal relationship of the independent variables and dependent variable due to the cross-sectional nature of the data.

## Conclusion

In this study, both individual level and community level factors were associated with having protected birth against NT. Mothers with primary and above education, having media exposure, not perceiving distance from the health facility as a big problem, having ANC visit, being women from communities with a higher level of women education, and a higher level of media exposure were significantly associated with higher odds of protected birth against NT. There were also geographical variations in the likelihood of protected births against NT. Therefore, strengthening maternal health services such as ANC visits and interventions related to increasing media campaigns regarding tetanus could increase the immunization against tetanus among reproductive-age women. In addition, it is also better to give attention to those reproductive age group women from remote areas and also better to distribute maternal services fairly and equally between regions.

## Acknowledgments

Special thanks go to the demographic and health survey program for granting us to access the data set for this study.

## Author Contributions

**Conceptualization:** Achamyeleh Birhanu Teshale, Getayeneh Antehunegn Tesema.

**Data curation:** Achamyeleh Birhanu Teshale, Getayeneh Antehunegn Tesema.

**Formal analysis:** Achamyeleh Birhanu Teshale, Getayeneh Antehunegn Tesema.

**Investigation:** Achamyeleh Birhanu Teshale, Getayeneh Antehunegn Tesema.

**Methodology:** Achamyeleh Birhanu Teshale, Getayeneh Antehunegn Tesema.

**Resources:** Achamyeleh Birhanu Teshale, Getayeneh Antehunegn Tesema.

**Software:** Achamyeleh Birhanu Teshale, Getayeneh Antehunegn Tesema.

**Validation:** Achamyeleh Birhanu Teshale, Getayeneh Antehunegn Tesema.

**Visualization:** Achamyeleh Birhanu Teshale, Getayeneh Antehunegn Tesema.

**Writing – original draft:** Achamyeleh Birhanu Teshale, Getayeneh Antehunegn Tesema.

**Writing – review & editing:** Achamyeleh Birhanu Teshale, Getayeneh Antehunegn Tesema.

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
