## [Decision Letter · Decision Letter 0]

14 Aug 2020

PONE-D-20-07884

Determinants of Births Protected Against Neonatal Tetanus in Ethiopia: A multilevel analysis using EDHS 2016 data.

PLOS ONE

Dear Dr. Teshale,

Thank you for submitting your manuscript to PLOS ONE. After careful consideration, we feel that it has merit but does not fully meet PLOS ONE’s publication criteria as it currently stands. Therefore, we invite you to submit a revised version of the manuscript that addresses the points raised during the review process.

We look forward to receiving your revised manuscript.

Kind regards,

Jai K Das

Academic Editor

PLOS ONE

Journal Requirements:

2.We note that you have indicated that data from this study are available upon request. PLOS only allows data to be available upon request if there are legal or ethical restrictions on sharing data publicly. For information on unacceptable data access restrictions, please see http://journals.plos.org/plosone/s/data-availability#loc-unacceptable-data-access-restrictions.

3. We note you have included a table to which you do not refer in the text of your manuscript. Please ensure that you refer to Table 3 in your text; if accepted, production will need this reference to link the reader to the Table.

Reviewers' comments:

Reviewer's Responses to Questions

**Comments to the Author**

1. Is the manuscript technically sound, and do the data support the conclusions?

Reviewer #1: Partly

Reviewer #2: Yes

2. Has the statistical analysis been performed appropriately and rigorously? 

Reviewer #1: No

Reviewer #2: Yes

3. Have the authors made all data underlying the findings in their manuscript fully available?

Reviewer #1: Yes

Reviewer #2: No

4. Is the manuscript presented in an intelligible fashion and written in standard English?

Reviewer #1: No

Reviewer #2: Yes

5. Review Comments to the Author

Reviewer #1: The paper addresses prevention of maternal and neonatal tetanus in Ethiopia. This paper does not add any novel knowledge to what is already known. The paper has a lot of grammatical and language errors and will need a lot of work before being considered for publication.

Introduction:

1. Please correct "Since most of the neonatal tetanus infections OCCUR due to poor sanitary conditions during delivery.."

2. Add some discussion around the routine TT vaccinations and special vaccine programs in the country.

3. It would be helpful if the authors could specify the latest coverage figures for ANC, skilled birth attendant and sanitary practices where they have specified "Even though there is low coverage of maternal health services such as antenatal care and skilled birth attendant delivery as well as poor sanitary practice during delivery in Ethiopia".

Methods:

1. Please correct "The individual-level factors used in this study WERE;..."

2. Please correct "..., and ever had of terminated pregnancy"

3. The authors state that "The community-level factors, community-level media exposure and community level of women's education, were created by aggregating individual-level factors since these variables are not directly found from the survey." It is not clear from the text that how these variables were aggregated and created. Moreover, if these variables are already considered at the individual level; I see no added benefit of constructing a new aggregate variable from individual data.

4. I would suggest if authors could provide some operational definitions for some variables in Table 1 including wealth status and media exposure.

Results:

1. The methods section specify that "... a total weighted sample of 7590 women who gave birth within two years preceding the survey was used" while in the results section it states that "A total weighted sample of 7590 women who gave birth within five years preceding the survey was used in our analysis." Please make this information consistent throughout the text.

2. Table 1 specifies that most of the women in the sample belonged to Oromia (41%); Amhara (21%) and SNNPR (21%) regions. It would be helpful if the authors could provide some details on these regions in the text.

3. As stated earlier, it has not been specified how were the community level aggregates for 'Community-level of women education' and 'Community-level of media exposure' were obtained. Please add these details and also specify the why these variables were generated.

4. It would be inappropriate to make conclusions about associations between religion and regions ("Regarding religion, mothers from Amhara and Afar had 41% [AOR = 0.59; 95%CI: 0.38,0.92] and 59% [AOR = 0.41; 95%CI: 0.25,0.66] lower odds of having births protected against NT as compared to mothers from Addis Ababa. Besides, mothers from Harari had 1.88 [AOR = 1.88; 95%CI: 1.15,3.07] times higher odds of having a protected birth against NT.") since 80% of the sample were from three regions only.

5. Table 3: Please present the findings from one final model only.

Discussion:

1. Considering that the data pertains to three regions namely: Oromia (41%); Amhara (21%) and SNNPR (21%) regions; there should be some discussion around the generalisability of these findings in the context of these regions.

2. It might not be appropriate to make conclusions about the regional associations ("There were also regional variations of births protected against NT in this study. Being mothers in the afar and Amhara region are less likely to have births protected against NT and being in the Harari region has higher odds of having births protected against NT.)

Reviewer #2: Major comments

1. In the background, where authors have made a case on TT vaccination coverage, it is important to mention the coverage of TT vaccination in women of reproductive ages in Ethiopia. At the same time, it is important to highlight the neonatal tetanus related deaths in the country?

2. Line 84-88: Authors have quoted predictors from Asian studies, it would be important to underscore the literature on TT from African context. Many of the studies are cited which are from Pakistan, Bangladesh and even from Cairo.

3. The current DHS from Ethiopia was published in 2019. However, authors have used 2016 DHS findings. Can you please explain the reason for not utilising the current survey?

4. Mother’s media exposure has been explained with studies from Africa and also with studies from Israel, and Indonesia which doesn’t look appropriate here. Authors should explain the predictors in light of studies from African region only. Again in line 225, distance as a predictor has been explained with study from Peshawar, Pakistan.

5. I believe the region specific variation needs further explanation.

Minor comments

6. It is better to use low and middle income countries, in place of developing countries.

7. Line 86, Antenatal care should be antenatal care

8. Line 89, antenatal care can be used as ANC as it has been abbreviated before.

9. Line 165: please write in full what is PCV?

10. Line 189: it should be region and not religion.

11. Line 197 and 198: this heading looks odd here?

6. PLOS authors have the option to publish the peer review history of their article (what does this mean?). If published, this will include your full peer review and any attached files.

Reviewer #1: No

Reviewer #2: No

---

## [Author Response · Author response to Decision Letter 0]

15 Sep 2020

Date: August 2020 

Point by point response to editor’s and reviewers comment

Title: Determinants of Births Protected Against Neonatal Tetanus in Ethiopia: A multilevel analysis using EDHS 2016 data.

Manuscript number: PONE-D-20-07884

Dear editor and reviewers: We really appreciate your useful/valuable comments and suggestions for improving this manuscript. Below is a point-by - point response to the questions / comments you raised. Thank you again for considering this manuscript.

Author’s response to editor’s comment 

Author’s response: Thank you. We amended the manuscript according to the journal style.

2. We note that you have indicated that data from this study are available upon request. In your revised cover letter, please address the following prompts:

b) If there are no restrictions, please upload the minimal anonymized data set necessary to replicate your study findings as either Supporting Information files or to a stable, public repository and provide us with the relevant URLs, DOIs, or accession numbers. 

 Author’s response: Thank you. In the revised cover letter, we stated that all result-based data is available in the manuscript. However, the data set was accessed through legal requesting and we cannot attach here with the manuscript as supporting information since it is not ethically acceptable. However, anyone who want the data set can access from the Measure DHS program at http://www.dhsprogram.com, through legal requesting. In addition, in the revised manuscript, we put the authorization letter, which stated, “The data must not be passed on to other researchers/bodies without the written consent of DHS. However, if you have co-researcher registered in your account for this research paper, you are authorized to share the data with them”, as supporting information. 

3. We note you have included a table to which you do not refer in the text of your manuscript. Please ensure that you refer to Table 3 in your text; if accepted, production will need this reference to link the reader to the Table.

Author’s response: Thank you. We refer/cite table 3 in the text. 

AUTHORS RESPONSE TO REVIEWERS COMMENT 

Reviewer #1 

The paper addresses prevention of maternal and neonatal tetanus in Ethiopia. This paper does not add any novel knowledge to what is already known. The paper has a lot of grammatical and language errors and will need a lot of work before being considered for publication.

Author’s response: Dear reviewer, thank you for an important issue you raised. This study is based on a nationally representative data, studies conducted before were either institution based or conducted in specific areas of the country, which might be very crucial for policymakers or other responsible bodies. In addition, all of the studies conducted before concerns about individual level factors but this study incorporates community level factors. Therefore, this study helps policymakers to set appropriate interventions at both individual and community level. 

Moreover, we extensively edited and corrected language and grammatical errors in our manuscript. 

Introduction:

1. Please correct "Since most of the neonatal tetanus infections OCCUR due to poor sanitary conditions during delivery.."

Author’s response: Thank you for the comment. We amended it in the revised manuscript.

2. Add some discussion around the routine TT vaccinations and special vaccine programs in the country.

Author’s response: Dear reviewer thank you for this important concern you raised. We incorporated the routine immunization/vaccine programs in Ethiopia (See the first paragraph of the background section). Regarding the routine TT immunizations in Ethiopia, we indicated that TT vaccine is part of maternal health services and every reproductive age women should receive five doses of TT (as early as possible, after 4 weeks of the 1st dose, after 6weaks of the 2nd dose, after 1years of the 3rd dose, and after one years of the 4th dose) (see background section paragraph five).

3. It would be helpful if the authors could specify the latest coverage figures for ANC, skilled birth attendant and sanitary practices where they have specified, "Even though there is low coverage of maternal health services such as antenatal care and skilled birth attendant delivery as well as poor sanitary practice during delivery in Ethiopia".

Author’s response: Thank you for your comment. We consider it/we added the figures (about ANC skilled birth attendant coverage) in the revised manuscript. However, it was difficult to have an actual figure regarding poor sanitary practices during delivery but in reality, there is inadequate sanitary practices during delivery in poor clinical setups like Ethiopia

Methods:

1. Please correct "The individual-level factors used in this study WERE;..."

Author’s response: Amended in the revised manuscript.

2. Please correct "..., and ever had of terminated pregnancy"

Author’s response: Amended accordingly

3. The authors state that "The community-level factors, community-level media exposure and community level of women's education, were created by aggregating individual-level factors since these variables are not directly found from the survey." It is not clear from the text that how these variables were aggregated and created. Moreover, if these variables are already considered at the individual level; I see no added benefit of constructing a new aggregate variable from individual data.

Author’s response: Thank you for raising this important issue. We indicated how these community level variables were created in the revised manuscript (see operational definitions). As you know community level factors are very important factors for utilization of maternal health services, we considered the community level factors by aggregating them from the respective individual level factors to indicate the neighboring effect. This helps policymakers to take intervention at both individual and community levels. For example, women from communities with lower level of media exposure might be clustered in specific areas and taking appropriate intervention in this group of women could have a great advantage to increase maternal health services including immunization services. 

4. I would suggest if authors could provide some operational definitions for some variables in Table 1 including wealth status and media exposure.

Author’s response: Thank you for the comment. We accepted the comment and put the operational definition of some important variables in the revised manuscript. 

Results:

1. The methods section specify that "... a total weighted sample of 7590 women who gave birth within two years preceding the survey was used" while in the results section it states that "A total weighted sample of 7590 women who gave birth within five years preceding the survey was used in our analysis." Please make this information consistent throughout the text.

Author’s response: Thank you. It was to mean “within five years” and amended it in the revised manuscript. 

2. Table 1 specifies that most of the women in the sample belonged to Oromia (41%); Amhara (21%) and SNNPR (21%) regions. It would be helpful if the authors could provide some details on these regions in the text.

Author’s response: We considered your comment in the revised manuscript (details about region is indicated).

3. As stated earlier, it has not been specified how were the community level aggregates for 'Community-level of women education' and 'Community-level of media exposure' were obtained. Please add these details and also specify the why these variables were generated.

Author’s response: Thank you. These variables were created since they were not directly available in the DHS but known to have associated with maternal health service utilization by different studies (such as a study by Yebyo HG, Gebreselassie MA, Kahsay AB; 2014). As we stated before these community level variables were created to show the effect of these variables at the cluster or the community level (see operational definition). Identifying factors at the community and individual levels could help policymakers to intervene both at individual and community levels.

4. It would be inappropriate to make conclusions about associations between religion and regions ("Regarding religion, mothers from Amhara and Afar had 41% [AOR = 0.59; 95%CI: 0.38,0.92] and 59% [AOR = 0.41; 95%CI: 0.25,0.66] lower odds of having births protected against NT as compared to mothers from Addis Ababa. Besides, mothers from Harari had 1.88 [AOR = 1.88; 95%CI: 1.15,3.07] times higher odds of having a protected birth against NT.") since 80% of the sample were from three regions only.

Author’s response: Really thank you for the comment. As you know, in the EDHS, to generate statistics that are representative of the Ethiopia as a whole (in the 11 regions), the number of women surveyed in each region should contribute to the size of the total sample in proportion to size of the region. However, if some regions have small populations, then a sample allocated in proportion to each region’s population may not include sufficient women from each region for analysis. To solve this, regions with small populations were oversampled. In addition, a sampling statistician determines how many women should be interviewed in each region in order to get reliable statistics. Furthermore, in order to get statistics that are representative of Ethiopia, the distribution of the women in the sample needs to be weighted (or mathematically adjusted) such that it resembles the true distribution in Ethiopia. Women from a small region, like Harari, Dire Dawa… should only contribute a small amount to the national total. Women from a large region, like Oromia, Amhara, and SNNPR should contribute much more. Therefore, DHS statisticians mathematically calculate a “weight” which is used to adjust the number of women from each region so that each region’s contribution to the total is proportional to the actual population of the region.

Therefore, since our analysis was based on weighting we expect the appropriate statistical estimate and representativeness of the sample in each region and the country in general. Due to this, we authors putt region as independent variable and make a conclusion for its association with our outcome variable. Dear reviewer we are open to remove it if it does not make sense for you still.

5. Table 3: Please present the findings from one final model only.

Author’s response: Dear reviewer thank you for the comment. As you indicated, we can put the final model only. However, since it was a multilevel analysis it is better to indicate all the models we fitted in the final table. This makes the paper consistent and easy to understand for readers since we indicated as we fitted four models in the method section as well as in the result section (random effect analysis). Moreover, papers on multilevel analysis report all the models fitted, not only the final model, in one table to make clear for both statisticians and other readers such as clinicians. If it does not make sense for you still, we are open to put the final model only (which includes both individual and community level variables simultaneously). 

Discussion:

1. Considering that the data pertains to three regions namely: Oromia (41%); Amhara (21%) and SNNPR (21%) regions; there should be some discussion around the generalizability of these findings in the context of these regions.

Author’s response: Thank you for raising very important issue again. As we stated above, the EDHS data was collected based on two-stage cluster sampling proportional to the total populations of the regions, in which the three regions had the highest total population, so had large samples. However, with sampling and weighting, it is possible to have/interview enough women to provide reliable statistics at national and regional levels. So, our analysis was based on weighting which restore representativeness of the samples. In addition, we employed a multilevel analysis in order to have better statistical estimate. Therefore, we can generalize our findings in the context of the regions of Ethiopia.

2. It might not be appropriate to make conclusions about the regional associations ("There were also regional variations of births protected against NT in this study. Being mothers in the afar and Amhara region are less likely to have births protected against NT and being in the Harari region has higher odds of having births protected against NT.)

Author’s response: Thank you. We amend it accordingly in the revised manuscript. We avoided “Being mothers in the afar and Amhara region are less likely to have births protected against NT and being in the Harari region has higher odds of having births protected against NT” and we put simply as there was regional variations of protected births against NT in general. Dear reviewer, as we sated before we can make conclusions about regional associations since we conducted our analysis based on weighting to make the sample representative to each regions and the country in general.

Reviewer #2

1. In the background, where authors have made a case on TT vaccination coverage, it is important to mention the coverage of TT vaccination in women of reproductive ages in Ethiopia. At the same time, it is important to highlight the neonatal tetanus related deaths in the country?

Author’s response: Dear reviewer thank you for raising an important issue. We accept your comment on highlighting the neonatal tetanus related deaths in the country. However, most NT deaths occur in the community and are not reported due to the population size (make it difficult for surveillance) and traditional value towards neonatal death, we are unable to find the current exact figure and highlight the neonatal tetanus related deaths in the country. Even though the country validated (partially) to have achieved NT elimination (reports less than 1 case per 1000 livebirths) this might be due to under reporting and we the authors expect more cases since most (greater than two-thirds) of mothers in Ethiopia gave birth at home. However, according to EDHS 2016 report, only half of reproductive women took sufficient doses of TT (two and above TT doses) during their pregnancy. Information about these are shown in the background section paragraph six. 

2. Line 84-88: Authors have quoted predictors from Asian studies, it would be important to underscore the literature on TT from African context. Many of the studies are cited which are from Pakistan, Bangladesh and even from Cairo.

Author’s response: Thank you. In the revised version of our manuscript we consider studies done in African countries. 

3. The current DHS from Ethiopia was published in 2019. However, authors have used 2016 DHS findings. Can you please explain the reason for not utilising the current survey?

Author’s response: Thank you for the comment. It is not fully available (only the mini EDHS released) so we used the DHS 2016 data. 

4. Mother’s media exposure has been explained with studies from Africa and also with studies from Israel, and Indonesia which doesn’t look appropriate here. Authors should explain the predictors in light of studies from African region only. Again in line 225, distance as a predictor has been explained with study from Peshawar, Pakistan.

Author’s response: Thank you. We consider only studies done in African context in the revised manuscript.

5. I believe the region specific variation needs further explanation.

Author’s response: Thank you. We consider your comment and add further explanations.

6. It is better to use low and middle income countries, in place of developing countries.

Author’s response: Thank you for the comment. We used low and middle-income countries instead of developing countries in the revised manuscript. 

7. Line 86; Antenatal care should be antenatal care

Author’s response: Amended in the revised manuscript

8. Line 89, antenatal care can be used as ANC as it has been abbreviated before.

Author’s response: Thank you. We consider it throughout the revised manuscript.

9. Line 165: please write in full what is PCV?

Author’s response: Thank you for the comment. We indicated its full meaning which means Proportional change in Variance in the method section like MOR and ICC. 

10. Line 189: it should be region and not religion.

Author’s response: Thank you. We amend it in the revised manuscript.

11. Line 197 and 198: this heading looks odd here?

Author’s response: Thank you. In plos one’s journal style, the title of the table should be putted immediately below the text expressing it. Therefore, it was the title of the table showing the text written above it and we added the table number in the text, which was missed.

---

## [Decision Letter · Decision Letter 1]

16 Nov 2020

Determinants of Births Protected Against Neonatal Tetanus in Ethiopia: A multilevel analysis using EDHS 2016 data.

PONE-D-20-07884R1

Dear Dr. Teshale,

We’re pleased to inform you that your manuscript has been judged scientifically suitable for publication and will be formally accepted for publication once it meets all outstanding technical requirements.

Kind regards,

Jai K Das

Academic Editor

PLOS ONE

Additional Editor Comments (optional):

Reviewers' comments:

Reviewer's Responses to Questions

**Comments to the Author**

1. If the authors have adequately addressed your comments raised in a previous round of review and you feel that this manuscript is now acceptable for publication, you may indicate that here to bypass the “Comments to the Author” section, enter your conflict of interest statement in the “Confidential to Editor” section, and submit your "Accept" recommendation.

Reviewer #2: All comments have been addressed

Reviewer #3: All comments have been addressed

2. Is the manuscript technically sound, and do the data support the conclusions?

Reviewer #2: Yes

Reviewer #3: Yes

3. Has the statistical analysis been performed appropriately and rigorously? 

Reviewer #2: Yes

Reviewer #3: Yes

4. Have the authors made all data underlying the findings in their manuscript fully available?

Reviewer #2: No

Reviewer #3: Yes

5. Is the manuscript presented in an intelligible fashion and written in standard English?

Reviewer #2: Yes

Reviewer #3: Yes

6. Review Comments to the Author

Reviewer #2: (No Response)

Reviewer #3: Thanks for responding and revising your manuscript. Please a few notes:

1. I would suggest if you could specify that both individual and community level variables were assessed in the abstract methodology.

2. There are still a grammatical corrections needed. Please read through and make changes accordingly.

7. PLOS authors have the option to publish the peer review history of their article (what does this mean?). If published, this will include your full peer review and any attached files.

Reviewer #2: No

Reviewer #3: **Yes: **Rehana A Salam

---

## [Editor Report · Acceptance letter]

18 Nov 2020

PONE-D-20-07884R1 

Determinants of Births Protected against Neonatal Tetanus in Ethiopia: a multilevel analysis using EDHS 2016 data. 

Dear Dr. Teshale:

I'm pleased to inform you that your manuscript has been deemed suitable for publication in PLOS ONE. Congratulations! Your manuscript is now with our production department. 

Kind regards, 

on behalf of

Dr. Jai K Das 

Academic Editor

PLOS ONE